# Symbolic Religious Landscape: Religious and Patriotic Symbolism in the Pilgrimage Centres in Poland

**Justyna Liro** *, **Izabela Sołjan and Elzbieta Bilska-Wodecka**

Faculty of Geography and Geology, Institute of Geography and Spatial Management, Jagiellonian University, 31-007 Kraków, Poland
* Correspondence: justyna.liro@uj.edu.pl

**Abstract:** Combining religious and national symbolism is not a new phenomenon. There are known examples of countries where there were or still are connections between nationality and religion. Pilgrimage centres are examples of symbolic religious landscapes based on the presence of the sacred. Such anthropogenic landscapes are a visible result of culture formed under the influence of religion, a special spiritual and often national heritage expressed through sacred objects, visual evidence of religiousness and, likewise, national identity. Here, we present a detailed analysis of religious and patriotic symbolism present in the largest pilgrimage centres in Poland. Additionally, the paper discusses a. the historical and socio-cultural conditions of the presence and significance of these elements in religious landscapes; b. the strong relationships between religiousness and the sense of national identity, and c. the resulting significant importance of pilgrimage centres in the development and consolidation of a sense of national identity. Symbolic elements in the analysed pilgrimage centres refer to both universal religious content and cults popular in the Roman Catholic Church, as well as to the identity of the analysed places. In addition to religious symbolism, national and patriotic symbols often occur in Polish sanctuaries. Their occurrence is historically conditioned and, to a large extent, results from the strong ties between religiousness and national identity.

**Keywords:** religious symbolism; patriotic symbolism; symbolic landscape; religious landscape; pilgrimage centres; national identity

## 1. Introduction

Travelling to holy places is one of the oldest religious practices in human history and occurs in almost all religions (Kong 1990; Margry 2008; Reader 2003). Old and new places of worship still attract pilgrims and tourists, and since the end of the 20th century, there has been a revival of travelling to these places (Digance 2003; Eade 2017; Reader 2014; Sołjan and Liro 2021, 2022). The importance of the relationship between religious phenomena and geographic space, especially those related to pilgrimage sites, have been emphasized by numerous authors (Cohen 1992; Eade and Sallnow 1991; Holloway and Vallins 2002). Pilgrimage centres were mentioned as one of the elements sacralising landscape. Papers dealing with religious symbolism, the sacralisation of religious landscape constitute the theoretical basis of the article (Bilska-Wodecka 2004, 2012; Boyer 2011; Bryce and Roberts 1996; Cosgrove 1985; Jones 1918; Hulme 2017; Holly 2014; Lilley 2000, 2004; Mâle 1984; Myga-Piątek 2012; Przybylska 2014; Whistler 2016).

A symbol can be understood in two ways, i.e., (1) as a concept, object, having one literal meaning and more hidden meanings; (2) a motif or a set of motives being a sign of deeper, hidden content, intended to suggest its existence (Kokosalakis 2001; Richards 1995; also according to *Encyclopaedia Britannica*). In a general sense, a symbol is considered to be an indicator of a certain state of affairs based on a convention adopted in a given community (Jones 1918; Sztompka 2002). It is defined as a means of interpersonal communication and the basis of community life in the cultural aspect. It is important to separate the

terms 'symbol' and 'sign', in which the former carries meaning and is not indifferent to the recipient (Sztompka 2002). Its content is constituted in the process of interpretation and its direct sensory and emotional perception by recipients (Sztompka 2002). Religion consists mainly of systems of symbols that build strong, lasting moods and motivations in the recipients, and their presence in religious landscapes is obvious and ubiquitous (Geertz 1998).

The concept of landscape symbolism is defined in various ways owing to the interdisciplinarity of this term. It can manifest itself in the presence of individual objects, in the relationships between buildings and their surroundings, as well as in the metaphors and allegories contained in architectural elements (Mâle 1984). The specificity of sanctuaries means that they can be perceived as a kind of symbolic landscape, consisting of the values and meanings of the objects present in them. They are saturated with an emotional charge, speaking through symbolism and through compositional connections. For a believer, they can be a source of emotions and feelings that determine the formation of relationships of belonging and attachment to a given place (Cohen 1992). The main purpose of the presence of religious symbolism in pilgrimage centres is the presentification of the deity, and thus the affecting of the reception of a place by the faithful, i.e., consolidating religious feelings, presenting important dogmas, truths of the faith, or facts (e.g., apparitions), as well as presenting people and objects surrounded by worship

The architectural design of a pilgrimage centre is an example of a religious landscape based on the presence of *the sacred*. Such anthropogenic landscape, being part of cultural landscape, is a visible product of culture formed under the influence of religion (Lilley 2000, 2004; Sołjan 2012). It is also a fragment of geographical space, the structure and functions of which were developed as a result of the interaction of natural processes and components with cultural (religious) factors (Bilska-Wodecka 2004; Myga-Piątek 2012). Religious landscape is defined as a specific spiritual and cultural heritage expressed by sacred objects, built at a specific time and in a specific area, which are visual testimonies of religious culture (Lilley 2000; Myga-Piątek 2012). The organization of objects in religious landscape results from the experiencing of *the sacred* and the implementation of religious practices, while the presence of objects, i.e., churches, chapels, crosses, and statues is a kind of sacralisation of landscape (Przybylska 2014). As a result of this process, the area becomes particularly distinguished by the objects and religious symbols appearing in it, as well as by the rituals performed, ensuring direct contact with *the sacred* (Bryce and Roberts 1996).

In the current state of research, owing to the connection with the content of this paper, it is also worth distinguishing publications describing the meaning of such concepts as sacred space, sacred place, and *the sacred*. Religious landscape is defined as the space in which *the sacred* manifests itself (Eliade 1991, 1999; Mâle 1984). Its reception is subjective since it depends on the religiousness of a human being. A non-religious person perceives space as homogeneous, while for a religious person landscape is varied owing to the presence of *the sacred* (deity, God) in it (Eliade 1999; Bilska-Wodecka 2003). In terms of religion, a holy place is a special place of grace that has attracted pilgrims for centuries (Leeuw 1933). In the concepts of sacred places, the most important axiom is the presence of *the sacred*, defined very differently in different religions. It is the presence of the sacred that makes a place considered sacred become a place of pilgrimage. M. Eliade (1999) defines a pilgrimage as a religiously motivated journey to the symbolic centre of the world-*axis mundi*, or its representation, i.e., a holy place, a place of *the sacred*, separated from the zone of *the profane*. According to V. Turner (1969), a pilgrimage belongs to the rites of passage and is of liminal nature for a pilgrim. Detached from everyday life, a pilgrim is on his or her way to direct contact with *the sacred*. The pilgrimage destination (centre) is usually there, so the pilgrim must make some journey to that place. Referring to the concept of V. Turner (1969) and M. Eliade (1999), Chidester and Linentha (1995) point out that a holy place does not necessarily have to be the opposite of *the profane* separated by a clear border, because the zone of *the sacred* is inextricably linked with social and environmental reality. They agree with G. Leeuw (1933) who assumes that sacred buildings are the result of

the influence of a holy place. In this approach, the erection of a sanctuary is understood as a consequence of the sanctity of a given section of landscape. In this approach, the establishment of a sanctuary is understood as a consequence of the sanctity of a given section of space. The need for contact with *the sacred* in holy places is one of the basic needs of a religious (believing) man–*homo religious* (Bilska-Wodecka 2012). In every religion, holy places are places of prayer, worship, meditation, times of internal transformation of the pilgrim. Among the distinguishing features of contemporary pilgrimage centres are the dominant religious function, the sacralisation of space giving it a new qualitative dimension, its real or symbolic separation as the zone of *the sacred*, a specific public space within which certain behaviours are required (Sołjan 2012; Sołjan and Liro 2021, 2022). Their architectural appearance and the symbolism found in them can help the visitor to be closer to *the sacred*, deepen their religiousness or spirituality, strengthen the sense of community, and create a friendly place for spiritual experiences.

The landscape of pilgrimage centres may undoubtedly be interpreted in symbolic terms. Both the architectural forms of objects and the organization of space, have numerous religious references. This symbolism gives them value, distinguishes them from the surrounding homogeneous, non-religious landscape. Apart from the material heritage expressed primarily through architecture and religious art, the semantic layer of sanctuaries also includes intangible (spiritual) heritage expressed through holidays, customs, and religious ceremonies taking place in their area. The presence of patriotic elements next to religious symbolism is characteristic for Polish pilgrimage centres. Both religious and patriotic symbolism are also intended to document and commemorate, among other things, the events, people, and other content important from the perspective of Catholic piety and national consciousness.

The article presents, a detailed discussion of religious and patriotic symbolism present in the largest pilgrimage centres in Poland today. The paper focused on three research aims. They included a. discussing the historical and socio-cultural conditions of the presence and significance of these elements in religious landscapes, b. presenting the strong relationships between religiousness and the sense of national identity, and the resulting significant importance of pilgrimage centres in the creation and consolidation of the sense of national identity and religiousness. Symbolic elements present in the studied pilgrimage centres refer both to universal religious content and cults popular in the Roman Catholic Church, as well as to the identity of a given centre. In addition to religious symbolism, Polish sanctuaries often feature national, patriotic symbols. Their presence is historically conditioned and, to a large extent, results from the strong ties between religiousness and national identity.

## 2. Materials and Methods

The research covered twelve largest pilgrimage centres of the Roman Catholic Church in Poland (Bardo Śląskie, Jasna Góra in Częstochowa, Gietrzwałd, Kalwaria Zebrzydowska, Kałków-Godów, the Sanctuary of Divine Mercy and the Sanctuary of St. John Paul II in Krakow, Licheń Stary, Niepokalanów, Tuchów, Wejherowo, and Zakopane-Krzeptówki). In order to diversify the research sample, the selection was made on the basis of the following criteria: (1) the pilgrimage centres are functioning today (there is a pilgrimage movement to them); (2) they come from different historical periods; (3) they are located in various settlement units; (4) they differ in the range of their impact (excluding local ones).

Among the sanctuaries covered by the research, three (Bardo Śląskie, Jasna Góra in Częstochowa, and Gietrzwałd) date back to as early as the Middle Ages. The oldest is the pilgrimage centre in Bardo Śląskie, where a Cistercian monastery was founded and the first church was built in the 13th century. Since the 14th century, the faithful have been making pilgrimages to it because of the cult of the statue of the Mother of God with the Child. The genesis of the Marian cult and pilgrimage movement at Jasna Góra in Częstochowa dates back to the 14th century. In Gietrzwałd, there was a church with an miracolous image of the Mother of God from the 15th century. A breakthrough event in the history of this centre

was the Marian apparitions in 1887. Another group of centres date back the 17th century. These are: Kalwaria Zebrzydowska, Wejherowo, and Tuchów. Their foundation is related to the post-Tridentine revival (17th–18th century), and thus a strong revival of religious life after the period of the Reformation. Marian apparitions, recorded in great numbers in the 19th century, also took place in Poland. Apart from Gietrzwałd, they significantly affected the sanctuary in Licheń Stary, being the basis of its establishment and development. Other pilgrimage centres were established in the 20th and 21st centuries. In the interwar period, on the initiative of Fr. Maximilian Maria Kolbe, Niepokalanów–a place of Marian cult was established. Kolbe's death in the KL Auschwitz concentration camp in 1941 during World War II initiated the development of the cult and the pilgrimage movement to Niepokalanów as a place associated with him. The genesis of the centre of the worship of Divine Mercy in Kraków-Łagiewniki dates back to the mid-1930s, and its gradual development has been going on since the late 1970s. The youngest two Marian sanctuaries in Kałków-Godów and Zakopane-Krzeptówki were founded in the second half of the 20th century, while the Sanctuary of St. John Paul II in Krakow at the end of the first decade of the 21st century (Figure 1).

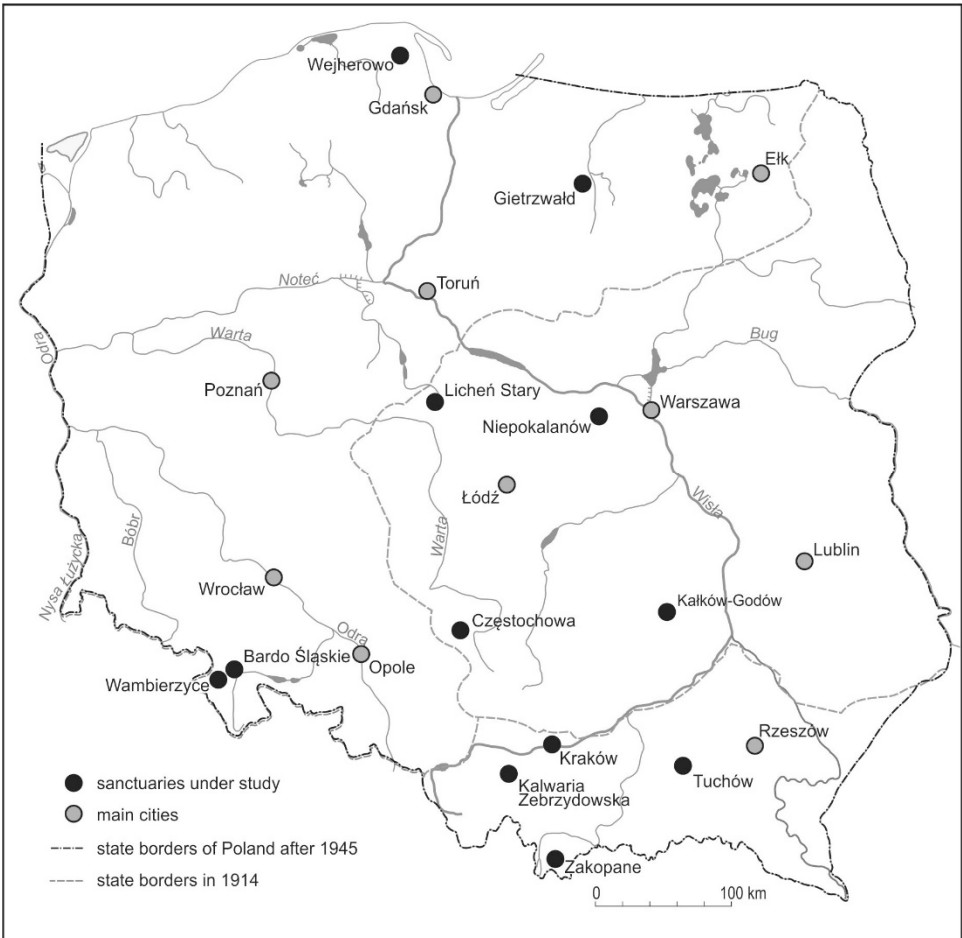

**Figure 1.** Location of the analysed sanctuaries within the historical borders of Poland. Source: Authors own study.

The presence of national symbols as well as plaques and monuments commemorating events from the history of Poland were assessed using qualitative methods-during the field inventory-by comparing the type and manner of commemoration. Their presence in a sacral building or in the sanctuary zone indicated the presence of a sense of national identity. Owing to the main object of worship, the examined sanctuaries can be divided into: Marian sanctuaries–seven centres; sanctuaries of saints–two centres (including Niepokalanów,

where the cult is combined with Marian cult, sanctuaries of our Lord–Kraków-Łagiewniki, and two Passion-Marian sanctuaries (Kalwaria Zebrzydowska and Wejherowo). In these sanctuaries, the passion cult associated with the Calvary chapels is combined with the Marian cult, while the worship of the Divine Mercy in Krakow-Łagiewniki is accompanied by the cult of St. Faustina Kowalska. Six of the analysed centres have an international range of impact, while another three supra-regional, and three regional.

The materials come from the observations and field inventory carried out in the years 2012–2022. The following methods were used in the article: archive and library query, field inventory, cartographic methods, and descriptive-analytical methods.

## 3. Results

### 3.1. Historical and Socio-Cultural Conditioning

Historical and socio-cultural conditioning has a significant impact on the current network of pilgrimage centres in Poland and their symbolic religious landscape. The analysis of the presence of national symbolism in the Roman Catholic Church in Poland and religious symbolism in the '*historical consciousness*' of Poles is present in the state of knowledge. Religious symbolism is analysed in the context of the European values (e.g., Mydłowska 2022) and as an important element of national identity (Mach 2007; Topidi 2019).

Christianity was an important factor in the formation and unification of medieval Europe. Along with the adoption of this religion, the process of sacralisation or resacralisation of the territory took place. New places of worship were created, sometimes the existing pre-Christian ones were transformed into Christian ones. Owing to religious, political, and socio-cultural factors, selected places often became pilgrimage centres popular with the faithful. In the 4th–5th centuries in Europe, the cult of the Holy Sepulchre, saints, and martyrs began to develop on a larger scale. An important element of the piety of that period was the belief in the intercession of saints, therefore, places associated with them grew in popularity, and making pilgrimages to them became a common religious practice in the Middle Ages.

Making pilgrimages in Poland has a centuries-old tradition, and the beginnings of the pilgrimage movement to Christian centres can be dated to the end of the 10th and the first half of the 11th century (Jackowski 1995). As in the whole of Europe, the centres of worship which contained relics or which were associated with the life or death of persons who had died in the opinion of sanctity developed to the greatest extent at that time. The church hierarchy and the rulers, who, thanks to bringing relics, had an influence on the establishment and development of a pilgrimage centre, played an exceptional role in the foundation of sanctuaries. Lord sanctuaries also played an important role in the Middle Ages. The first centres of this type in Poland were established in the 12th–14th centuries as a result of the crusades to the Holy Land. In the first centuries of the presence of Christianity in Poland, the cult of hermits also developed. The beginnings of the Marian cult in Christianity are related to the Council of Ephesus in 431, during which the dogma granting Mary the status of the Mother of God was proclaimed. The Marian cult in Poland appeared together with its baptism in AD 966 and played a significant role in the history of the country, many times integrating society and consolidating the feeling of national unity. This cult began to gain more and more importance from the 14th century. Founded in 1382, the monastery on Jasna Góra played an important role in this matter. In 1384, a picture of the Black Madonna was placed in the chapel. Soon the place became a famous centre of worship, which the faithful began to make pilgrimages to.

The Reformation and the anti-church activities in the 16th century had a negative impact on the development of pilgrimage centres in Europe. The response to the Reformation movements was the Council of Trent (1545–1563) and the Counter-Reformation aimed at the renewing of religious life in the Roman Catholic Church. The post-Tridentine reform resulted, inter alia, in the emergence of new sanctuaries, especially those associated with miraculous Marian images, and pilgrimage centres based on Calvary establishments.

The greatest intensity of Calvary foundations was observed from the beginning of the 17th century to the middle of the 18th century. Owing to the nature of Polish religiousness and the enormous importance of the Marian cult at that time, images of the Mother of God were often placed in Calvary sanctuaries. The development of Marian piety resulted in an increase in the popularity of a given Calvary and a revival of the pilgrimage movement. However, Calvary sanctuaries in the Roman Catholic Church in Poland have never achieved such popularity as Marian cult centres (Bilska-Wodecka 2003).

The aforementioned Council of Trent consolidated the cult of miraculous images in the Roman Catholic Church. This contributed to the increase in the popularity of places with miracle-working paintings or statues, and their cult in the 17th and 18th centuries developed on an unprecedented scale. In addition to Reformation activities, the difficult political situation in Poland at that time, marked by wars, affected the special Marian piety in the Baroque period. The defense of the monastery at Jasna Góra in Częstochowa against the Swedish army in 1655 and the Lviv Oath of King John II Casimir in 1656, in which the king proclaimed the Mother of God Queen of Poland, further consolidated the priority position of this centre in the country (Jackowski 1995). As a result of those events, the Jasna Góra sanctuary began to be strongly associated with national content, and for centuries the cult of Our Lady of Częstochowa has had a strong influence on the religiousness of Poles, while pilgrimages to this centre have played an important role in integrating society to the present day (Mach 1996).

The difficult socio-political situation at the end of the 18th century related to the partitions of the country had a negative impact on the functioning of the existing pilgrimage centres and on the establishment of new ones. The partitioning states (Prussia, Austria, and Russia) aimed at the gradual liquidation of the national consciousness of Polish society. During the partitions, the society's attachment to faith, in which Poles found an opportunity to express their patriotic feelings, got consolidated. The popularity of religious practices, including pilgrimages, was a characteristic phenomenon. In addition to religious content, they also contained patriotic elements and often constituted a kind of demonstrations against the partitioners, while the pilgrimage centres contributed to the consolidation of religious and national ties. The so-called national pilgrimage movement developed, while places related to the history of the country aroused great interest again. Jasna Góra in Częstochowa played a special role, becoming a symbol of the unity of a divided society.

Particular attention should be paid to the second half of the 19th century. At that time, the adoption of the dogma of the Immaculate Conception of the Blessed Virgin Mary in 1854, as well as the foundations of the so-called apparition sanctuaries in Europe, i.e., Paris (1830), La Salette (1846), Lourdes (1858), and later at Fátima (1917), contributed to the strong development of Marian piety. Also, the largest pilgrimage centres of the 19th century in Poland were still associated with the cult of the Virgin Mary, which additionally gained popularity thanks to the apparitions recorded, among other things, in Licheń Stary (1850–1852), Wiktorówki (near Zakopane 1861), Gietrzwałd (1877), and Szczyrk (1894).

The re-development of the sanctuaries took place in the first years after the end of World War II. During the communist period, the state authorities took many actions that made their functioning difficult. As during the partitions, pilgrimage centres played an important unifying role, supporting the initiatives of activists opposing the policy of the communist party and the government. After the period of Stalinist restrictions (1947–1956), they enjoyed a greater freedom of action, which was manifested, among other things, by important religious actions initiated by church authorities. Starting from the 1960s, the coronations of Marian images were one of the most important manifestations of religious life in Poland. During the entire post-war period, the cult of the Virgin Mary developed strongly. In this regard, the leading role was played by the Jasna Góra sanctuary (Częstochowa), to which walking pilgrimages have been made since the 1960s (approx. 140,000 people/year), covering the entire country. The most pilgrims, i.e., about 1,6 million, visited this centre during the World Youth Day in 1991.

Since the 2nd half of the 20th century and in the 21st century there has also been a noticeable revival of the cult of saints and the blessed, and the foundations of sanctuaries dedicated to them have become more and more frequent. The cult of Divine Mercy developed also in the second half of the 20th century. The most important trends in the pilgrimage movement in Poland in the 21st century include the renaissance of making pilgrimages to Santiago de Compostela.

Nowadays, in Poland, where the phenomenon of making pilgrimages has a centuries-old tradition (Jackowski 1995), there are now nearly 800 sanctuaries and places of special worship of the Roman Catholic Church. Marian sanctuaries (over 500) are definitely dominant, including 230 with crowned images. Nearly 150 sanctuaries are centres dedicated to saints and the blessed, and 100 are associated with various forms of the cult of Our Lord (Mróz 2019). The Sanctuary of Our Lady of Częstochowa at Jasna Góra (Częstochowa) is the most important sanctuary of international significance. Among the most frequently visited, the following should also be mentioned: the Sanctuary of Divine Mercy in Kraków-Łagiewniki, the Marian Sanctuary in Licheń Stary, the Passion-Marian Sanctuary in Kalwaria Zebrzydowska, and the Sanctuary of St. Maximilian Maria Kolbe in Niepokalanów.

### 3.2. Religious Symbolism in Pilgrimage Centres in Poland

Religious symbolism is present in all centres of worship. Also, the one found in sanctuaries expresses and, to some extent, presentifies *the sacred* (Ferguson 1961). It is through its content that the symbol introduces believers to the world of *the sacred*. Material elements are sensually accessible, while symbols indicate the spiritual realm. Each temple, being a sign of the presence of *the sacred* in the earthly reality, is the centre of the universe. Referring directly to Catholicism, the Eucharist is celebrated and sacraments are administered in churches. Churches and other sacred buildings, together with the rites held in them, have a transcendent dimension (Hulme 2017). They become holy places, separated from secular reality, where the community of the faithful can gather and which are dedicated to the conduct of worship.

When analysing the religious symbolism of the studied sanctuaries, attention should be paid to the fact that there is both the symbolism typical for all Catholic objects of worship as well as that associated only with a given sanctuary or a group of sanctuaries promoting a similar cult (e.g., Our Lady of Fátima, Divine Mercy, etc.).

Since the 4th century on, the cross has had a universal dimension among all Christian symbols. It is the most common symbol in Christian culture. It was usually the first element of the sacralisation of landscape. Crosses are also often separate objects. In 2000, the so-called millennium crosses were founded at temples across the country. Such objects are located, among other things, in the three analysed centres in Gietrzwałd, Kałków-Godów, and Tuchów. In Catholicism, the Host also plays a leading role, symbolizing the mystical Body of Christ. These two symbols are present in all Catholic centres, organizing space and endowing it with special value. This order in sanctuaries overlaps with another one, connected with a specific object of worship, especially venerated at a given centre. Theologically, Eucharistic worship is of the utmost importance, but the behaviour of pilgrims does not always attest to this. Queues in front of miraculous images, tombs, or relics of saints, or the main services held in front of them confirm the above statement. It is enough to give the example of Jasna Góra, where the faithful usually direct their first steps to the miraculous icon of the Mother of God. In the landscape of the analysed pilgrimage centres, symbolism associated with a given centre in terms of identity is clearly visible.

The symbolic value of objects of worship especially venerated in sanctuaries results neither from the theological analysis of a given image, nor from its artistic properties, but above all from the exceptional events attributed to them, which were observed as a result of them, and from the relationship with this particular place. It is a bit different in the case of sanctuaries dedicated to saints. It is their heroic deeds and holiness that are important here, and the elements of the extraordinary and of the miraculous must be present in them,

because the specificity and peculiarity of sanctuaries consists in it. The founding myth of a centre, the place or object of worship connected with it lie at the foundation of the symbolic landscape created in sanctuaries. Particular objects or places of worship are of extraordinary value in themselves. The narrative assigned to them is also their essence. It is obvious that the symbolism contained in a given image or statue translates into pastoral activities and forms of piety developed in a pilgrimage centre. In the main Marian centres, the strongest emphasis is placed on elements related to the cult of the Mother of God, e.g., at Jasna Góra (Częstochowa), Bardo Śląskie, Gietrzwałd, Licheń Stary, Kałków-Godów, or Zakopane-Krzeptówki. On the other hand, in the Sanctuary of Divine Mercy in Kraków-Łagiewniki, the greatest attention is paid to the image of Merciful Jesus and the relics of St. Faustina, and in the nearby centre of St. John Paul II to the symbolic elements referring to him.

An important factor forming symbolic landscape is also the message flowing from the centre and the history of the sanctuary. Important events and distinguished persons are thus "recorded" in the landscape and collective memory. The figures of visionaries, priests, bishops, and popes contributing to the development of the cult propagated in a given centre are often immortalized. For example, in Gietrzwałd, two single Marian chapels commemorate the apparitions of 1877. In Licheń Stary, symbolic graves-monuments to the witnesses of the apparitions of Mikołaj Sikatko (1787–1857) and Tomasz Kłossowski (1780–1848) were erected, as well as a grave of and a monument to Bishop Roman Andrzejewski (1938–2003), the national chaplain of farmers, who often came to this centre with pilgrimages of rural communities. Bishop Leon Wałęga (1859–1933), the guardian of the sanctuary in the 1930s, was buried at the main temple in Tuchów. Since 1859, at Jasna Góra, there has been a monument to Fr. August Kordecki (1603–1673), the prior of the monastery and commander of its defense during the Swedish Deluge in the 17th century. Not only did the defense of the monastery against the Swedish army in 1655 initiate the cult of Our Lady of Częstochowa as the Queen of Poland, but it also became a permanent and important element of Polish history and religiousness. Similarly, the inclusion in the Jasna Góra landscape of the statue of the Primate of the Millennium, Cardinal Stefan Wyszyński (1901–1981) in 1997, symbolically shows his connections with this sanctuary and the role of the centre in difficult communist times.

In many cases, valuation in pilgrimage centres is emphasized by the architectural forms of objects and compositions of the establishments, which certainly also have a symbolic interpretation. For centuries, temples in which venerated objects of worship were kept, have been distinguished by their monumentality (e.g., medieval cathedrals), beauty, and richness of the interior (baroque churches). Also on a regional scale, these were usually better equipped churches. It is enough to mention here the Marian basilicas in Bardo Śląskie, Gietrzwałd, Kalwaria Zebrzydowska, and Tuchów. Also, the composition of pilgrimage centre establishments usually symbolically emphasizes the importance of the main temple. Usually, it is the axis of the entire establishment. Only since the time of the construction of the sanctuary in Lourdes have significant changes occurred in this regard, and nowadays the architectural dominant of the complex is not always the dominant in the religious sense, which in Lourdes is not the basilica, but the grotto. The situation is similar at the Sanctuary of Divine Mercy in Kraków, where the most important objects of worship remained in a small convent chapel, and the Basilica of Divine Mercy was consecrated nearby in 2002.

However, in all analysed pilgrimage centres, regardless of their leading type, Marian, Christological, and hagiographic symbolism is accompanied by other elements. In Marian sanctuaries there are chapels, statues, and smaller buildings dedicated to the worship of Jesus, or saints and the blessed. Likewise, there are numerous references to the cult of the Mother of God in Passion centres. Thus, typical cults and forms of piety (especially popular piety), inscribed in the tradition of the Church, find their expression regardless of the type of sanctuary.

The universal and timeless cult of the Virgin Mary manifests itself in the foundation of objects dedicated to the Mother of God. The exceptionally strong development of this cult in Poland contributes to the greatest number of representations and dedications (namings) not only of the main churches, but also of chapels (including rosary chapels), statues, and oratories occurring in sanctuaries. The time range of the foundation of these facilities is very extensive, from the times of the establishment of a given centre to the present day. In six of the twelve analysed centres there are rosary chapels with content related to the cult of the Virgin Mary. Their foundation in Gietrzwałd, Kałków-Godów, Licheń Stary, Tuchów, and Jasna Góra took place in the 1960s, and lasted until the beginning of the 21st century. On the other hand, in Bardo Śląskie, rosary chapels have been built since 1904. Of the planned 17 chapels, 3 are still missing.

### 3.2.1. Symbolism Related to the Marian Cult

The Marian cult was officially approved in the Church in AD 431 during the Council of Ephesus. However, in the following centuries, its content was deepened and new Mariological dogmas were formulated, which also translated into Marian symbolism and the ways of representing the Mother of God. Under the influence of the greatest Marian apparitions, the messages and images of the Virgin Mary referring to them became known throughout the Church, especially that of Our Lady of Lourdes, Fátima, and La Salette. In the group of the analysed pilgrimage centres it is visible in the case of a few. The sanctuary in Zakopane-Krzeptówki, established and functioning as the sanctuary of Our Lady of Fátima and of St. John Paul II, is brought to the forefront here. The entire symbolism of the centre revolves around the Pope's relationship with Fátima, especially with the so-called Third Secret of Fátima. The temple in Krzeptówki was built in the years 1987–1992 as a votive offering for the saving of John Paul II during the attempt on his life on 13 May 1981 and for his pontificate. In its central place above the altar there is a copy of the Marian figure from the Portuguese sanctuary. This centre has been in close contact with the Fátima sanctuary for many years and has spread its message. The Fátima message also inspired the establishment of the oratories in Licheń Stary and Kałków-Godów. It should be remembered that the message of Fátima, apart from its religious part, contained important political content related to the communist times. Mary announced the fall of communism, which is why the communist authorities in Poland hindered the development of this cult. The situation changed only in the 1980s, especially in the second half of that decade. At that time, oratories were built in Licheń Stary (1986) and Kałków-Godów (1989). They are to symbolize not only the victory of Mary, but also to emphasize the similarity of the messages in Licheń Stary and Fátima, especially the call to repent and do penance. In Lourdes, the symbol most often associated with the message of Virgin Mary is the grotto where Mary appeared to Bernadette Soubirous, and the statue of Our Lady Immaculate. Under the influence of Lourdes was St. Maximilian M. Kolbe. The events that had taken place in this centre affected his decision to found Niepokalanów, the name of which symbolically refers to the French sanctuary (*niepokalana* in Polish is *immaculate*). In 1929, he placed a statue of Our Lady of Lourdes in the church in Niepokalanów. Apart from Niepokalanów, the influence of Lourdes is visible in Kałków-Godów, where the Lourdes grotto was built in 1999. In addition, in the sanctuary of John Paul II in Krakow and in Zakopane-Krzeptówki, Lourdes is featured in the paintings showing the most important sanctuaries visited by the Polish Pope.

### 3.2.2. Symbolism Related to the Christological Cult

The cult of Our Lord manifests itself in sanctuaries, mainly in the form of the Passion cult, the presence of chapels and statues, most often included in the Stations of the Cross. The foundation of these greatest establishments fell on the times of popularity of Calvaries, i.e., in the 17th century. The Stations of the Cross are also founded in pilgrimage centres today in the form of chapels located in the landscape. The same cult was the factor which inspired the construction of the so-called Golgothas in Licheń Stary and Kałków-Godów

(1970s and 1980s). In these centres, however, they are not typical Passion presentations, recreating the path of Jesus' passion, because they are associated with numerous tragic events in the history of Poland, which is a phenomenon not found in other Catholic sanctuaries. In the last two decades, the image of Merciful Jesus has become an important symbolic element. It was affected by the development of the worship of Divine Mercy in the forms proposed by St. Faustina and the development of the centre in Kraków-Łagiewniki. Like the apparitions in Lourdes or Fátima, the message of Christ known from the writings of St. Faustina is universal and global. Although the first image of Merciful Jesus, painted in 1934 according to the diary of St. Faustina, is located in Vilnius, it is Łagiewniki that has become the main centre of this cult. Along the road leading to the chapel, there are boards with flags of various countries and the phrase "Jesus, I trust in you" translated into numerous languages. It symbolically emphasizes the universality of the cult and the opening of the sanctuary to all pilgrims, regardless of nationality and religion. The architectural idea that gives the basilica the shape of a boat can be read similarly as a place of refuge for those in need and seeking God's presence in the contemporary, increasingly secularized world.

### 3.2.3. Symbolism Related to the Cult of Saints and the Blessed

The representations of saints and the blessed are popular in the analysed pilgrimage centres. After the death of people living in the opinion of sanctity, then beatified and canonized, their cult develops, which entails the foundation of different sacral objects. Since the 2nd half of the 20th century, a clear revival of and an increase in the popularity of the cult of saints has been observed in Poland. This fact was affected, among other things, by the pontificate of Pope John Paul II, during which he declared 1341 persons blessed, including 154 from Poland, and 483 persons saints including 9 from Poland, the largest number of all popes to date. Since the beatification (1971) and canonization (1982) of the Franciscan St. Fr. Maximilian Maria Kolbe (1894–1941), churches, chapels, and statues dedicated to him have been founded. This fact decided also about the development of the sanctuary in Niepokalanów as a centre of the religious worship of the monk. Today, despite the promotion of the Marian cult, the sanctuary is associated primarily with St. Maximilian. St. Maximilian is presented as a symbol of a man who, in an inhuman place which was the German extermination camp in Auschwitz, selflessly gave his life for a fellow prisoner. In 1971, the chapel of St. Maximilian was erected in Kałków-Godów, and in 1979, in Licheń Stary, a mausoleum dedicated to his memory was founded. The figure of another Polish martyr, Bl. pr. Jerzy Popiełuszko (1947–1984), the chaplain of "Solidarity", murdered by the Security Service, was also immortalized in the two latter. Pr. Popiełuszko is a character-symbol of the fight against the communist regime in Poland in the 1980s. That is why, oratories dedicated to him were established in the centres of cult that displayed patriotic content as strongly as Licheń Stary and Kałków-Godów. The cult of St. Faustina (1905–1938)–nun in a convent of the Congregation of Sisters of Our Lady of Mercy–in Polish churches has been developing either as accompanying the worship of Merciful Jesus, or on its own. In the analysed centres, apart from Łagiewniki, the relics and images of this saint can be found in three sanctuaries (Jasna Góra, Licheń, Kałków-Godów). However, the most popular Polish saint in the last few decades has been John Paul II (1920–2005). His cult began to develop while the Pope was still alive. He was commemorated in very different ways in the public space. Therefore, there are streets, schools, hospitals, an airport named after him and about 800 monuments and 3 sanctuaries. The dynamics of its commemoration was so significant that there was talk of 'John-Paul-the-Second-isation' of cultural landscape in Poland (Przybylska and Sołjan 2015).

His monuments began to be founded at the beginning of the 1980s, and the greatest intensification of this process took place after his death (2005), beatification (2011), and canonization (2014). The first monument was erected in 1980 in the courtyard of the Bishop's Palace in Krakow. It is estimated that currently there are from 700 to 800 such objects in Poland. Most of them are located in the vicinity of temples. The process of

sacralisation of landscape by erecting churches dedicated to John Paul II, as well as smaller architectural objects, especially monuments, is present throughout the country (see also Brzozowski 2013). Of the twelve studied sanctuaries, papal monuments were erected in eight of them. The saint's images are present in all of the analysed centres. A unique accumulation of objects related to the person of John Paul II occurs in sanctuaries dedicated to him in Zakopane-Krzeptówki and in Kraków. An interesting example of representations related to his pontificate is also the Marian chapels in the Zakopane sanctuary and in the cult centre in Krakow. They differ in their architectural form, i.e., in Zakopane they are wooden chapels located in open space in Fátima Park, while in Krakow oratories occupying the lower part of the main temple. They are linked by the content of the representations, i.e., they commemorate the Marian centres which the Pope was closely associated with and which were visited by him.

As intended by the architect, the John Paul II Centre in Krakow has a symbolic dimension commemorating the places where Karol Wojtyła lived (see also Niedźwiedź 2017). The square in front of the main entrance to the temple was based on the model of the market square in Wadowice, the town where he was born. Narrow streets and courtyards between individual buildings symbolically reflect the architecture of Rome. Also, the natural materials used in the façades of the buildings refer to the places in Krakow that played an important role in the life of the saint, among others Wawel Cathedral, Collegium Maius in Jagiellonian Uniwesity, or St. Mary's Basilica. In addition, the walls of the lower church in the sanctuary are decorated with paintings depicting the Pope's visits to the largest Marian centres in Europe. A Priestly Chapel was built next to it, designed in the style of the crypt of St. Leonard inside Wawel Cathedral, where in 1946 pr. Karol Wojtyła celebrated his first mass. There is a plate there from the papal tomb in St. Peter's Basilica in Rome.

Among the saints in the analysed sanctuaries, there are symbolic elements related to the Capuchin St. Fr. Pio (1887–1968), who is one of the most famous and revered saints today; every year several million pilgrims come to his tomb in San Giovanni Rotondo in Italy. Among other saints, the most common are the statues of the patrons saint of the religious congregations that look after a sanctuary.

Summing up, in the analysed sanctuaries, religious symbolism is associated primarily with the nature of a given centre, the main object of worship, and the proclaimed message. The other elements refer to the forms of worship present in the entire Church or to persons and events significant from the point of view of a given centre. In most pilgrimage centres, this symbolism creates a relatively coherent system, consistent with the diagram presented above. Although this cannot be stated in the case of Licheń Stary and Kałków-Godów, the symbolism associated with the miraculous image in the foreground is lost in them among the numerous other symbols referring to various religious, patriotic, and historical themes. It gives the impression of certain randomness, an accumulation of many symbolic objects, in which the main idea of these sanctuaries has been lost.

### 3.3. Patriotic Symbolism in Pilgrimage Centres in Poland

The presence of objects (i.e., chapels, statues, epitaphs) containing, apart from religious symbolism, patriotic content is characteristic for pilgrimage centres in Poland. Patriotism, in common terms, means the loving of one's homeland as a place of origin or residence. Functioning as an element of a nation's culture, it is perceived as a social value. Elements of national symbolism have been present in centres of worship for centuries, but since the second half of the 20th century they have begun to be introduced more and more often into pilgrimage centres in Poland. All this had an impact on the creation of sanctuaries, which are mentioned in the literature as having a national character (Niedźwiedź 2010, 2014).

The simultaneous presence of religious and patriotic symbolism in sanctuaries in Poland results from historical conditioning. Such strong ties between national identity and religious consciousness are not observed in other European countries. The timelessness of the coexistence of the patriotic trend and the religiousness of Poles is noteworthy. It

was especially noticeable in times of increasing political, social, and economic repression. For centuries, pilgrimage centres have been not only places of prayer, but also a source of national consciousness and patriotic attitudes. Apart from fulfilling religious functions, they integrated the nation in times of partitions (1795–1918), uprisings, wars, and the communist rule (1945–1989). During those periods, participation in services and pilgrimages was a kind of act of manifestation or a symbol of identifying religiousness with national identity. Catholicism has become entrenched in culture, combining elements of Polishness and Christianity, while the common faith and belonging to the Church integrated diverse social groups.

The unique role of catholic pilgrimage centres in maintaining national identity is primarily affected by the strong and popular Marian cult in Polish society (de Busser and Niedźwiedź 2009). Marian elements, most often images of Our Lady of Częstochowa, were placed on the banners of Polish troops fighting in battles. The Marian cult and the pilgrimage movement associated with the Jasna Góra sanctuary have been performing important cultural and moral functions for centuries, while the centre itself is a symbol of national identity and unity for Poles. The patriotic trend in the pilgrimage movement was also strongly marked during the partitions. The loss of state sovereignty was associated with the risk of losing national and religious identity. Making pilgrimages was largely difficult or completely forbidden. The attachment of Poles to the Roman Catholic Church was a cause of concern for the communist authorities. They imposed numerous restrictions, seeking to limit the role of faith in the lives of the citizens of the socialist state. For this purpose, the Church and church services were under surveillance, it was forbidden to gather during ceremonies, organize pilgrimages, while participants in the services were harassed.

These activities also affected sanctuaries and the pilgrimage movement. Counter-initiatives included primarily religious ceremonies and acts addressed to the nation and emphasizing Jasna Góra as the spiritual capital of Poland, i.e., the celebration of the Millennium of the Baptism of Poland in 1966 or the coronation of the images of the Mother of God. At that time, Marian piety and the role of sanctuaries in the religiousness of Polish society got strongly consolidated. As during the partitions, in the communist times, pilgrimage centres played not only religious roles, but also those integrating society. Actions taken by the communist authorities were often ineffective and made Poles even more religious. At that time, content not approved by the authorities was disseminated in sanctuaries, including information about the Fátima apparitions and the cult of the Immaculate Heart of the Blessed Virgin Mary. The election of Karol Wojtyła as pope in 1978 strengthened the sense of value and awareness of one's own tradition. It is emphasized that this fact triggered the emergence of the "Solidarity" social movement opposing the ruling totalitarian system. Pilgrimages, especially to Marian sanctuaries in the 1970s and 1980s, constituted an important factor in the integration of society. The pilgrimages in which individual and religious motivations were combined with national ones had a special character. The activity of the church in the country is still often aimed at merging individual religiousness with patriotic feelings.

Among the objects and elements in pilgrimage centres combining religious and patriotic symbolism certain groups can be distinguished, taking into account the historical events they refer to and the time of their foundation. Before World War II, Jasna Góra was the main place of national remembrance owing to historical and social conditions. Compared to other centres covered by the study, references to national content were the most visible there. Until the end of the 18th century, pilgrims had been provided with access to the Treasury and the Knights' Hall with collections presenting the history of the place. Currently, the space of this centre is still strongly marked with symbolic content combining religious and national elements, and this place links national ideology with Catholic religion. Nowadays, the greatest intensity of the occurrence of national symbols can also be noticed in Licheń and Kałków-Godów. In these sanctuaries there are numerous patriotic, political, and military symbols expressed by national coats of arms and flags combined

with crosses and Marian symbolism. They are placed both in specially designated facilities (e.g., chapels, museum halls) and places (e.g., commemorative walls). In addition, in the sacred park in Licheń Stary, numerous patriotic monuments were founded to document and commemorate people who played an important role in the history of the country and the Church.

During the communist era, the authorities had a negative attitude towards any national-patriotic activities. The foundation of new objects, as well as the organization of religious services and events of this nature, were often banned and impeded, while participants in them were subject to repressions. The fall of the communist rule in the late 1980s affected the more and more frequent foundation of objects with national content. After the political transformation, a certain revival and intensification of founding them could be observed, and in the 1990s elements with patriotic symbolism were placed in Polish sanctuaries the most intensively. At that time, chapels, statues, and monuments were built commemorating, among other things, insurgents during the partitions, soldiers of the Home Army, victims of World War I and II, incl. political prisoners, repressed soldiers, or Siberians. Owing to their symbolism, unique objects are the collections of epitaphs, plaques, and oratories, such as Golgotha in Kałków-Godów, the National Remembrance Wall in Licheń Stary, and the National Remembrance Chapel at Jasna Góra. Inside the Golgotha Martyrology of the Polish Nation in Kałków-Godów there are several dozen chapels and oratories commemorating the tragic events in the history of the country. The so-called National Remembrance Wall in Licheń Stary performs a similar documenting role. In the 1990s, elements commemorating the activity of Solidarity were also founded. Monuments and epitaphs devoted to this organization were created in Licheń Stary, Kałków-Godów, and Jasna Góra. Another trend that emerged in the 21st century is the foundations that are aimed at the preservation of the memory of the victims of the 2010 air crash near Smolensk. In many sanctuaries plaques and epitaphs commemorating this event have been placed. In Licheń Stary, this event is also commemorated by the avenue of the victims of the Smolensk air crash, next to which there is a monument and 96 oaks planted there in 2011 (this number corresponds to the number of the victims). In 2012, in Kałków-Godów, in the vicinity of the Stations of the Cross and Golgotha, an epitaph and a monument, i.e., a concrete model of the government plane with photographs of some of the victims, were built. On the anniversary of the air crash, religious and patriotic services are held at the above-mentioned objects.

Apart from individual objects, such as chapels, statues, and epitaphs, in some of the analysed pilgrimage centres there are cultural objects housing memorabilia and other elements with patriotic content. Their creation is characteristic of the 21st century. Such functions have been performed by the National Memory Treasury at Jasna Góra (Częstochowa) since 2006, while the Museum of pr. Józef Jarzębowski (1897–1964) in Licheń Stary houses one of the largest collections of memorabilia of the heroes of the January Uprising (1863–1864), prisoners of Soviet labour camps and Nazi concentration camps in Poland.

In the other studied sanctuaries there are also symbolic references to patriotic content, however, they are of the nature of individual objects.

## 4. Conclusions

The article presents symbolic religious landscape by analysing the religious and patriotic symbolism present in the pilgrimage centres of the Roman Catholic Church in Poland. The analysis conducted in this respect revealed two basic trends in this regard. First of all, in the sanctuaries in Poland, like in other Catholic pilgrimage centres, religious symbolism most often associated with a specific sanctuary and a Eucharistic or Marian cult, as well as with the cult of saints common in the Church, occurs. In the latter group, the most common examples are the saints and the blessed who lived in Poland and Europe in the 20th and 21st centuries, and there are especially numerous references to Pope John Paul II. However, another tendency that distinguishes Polish pilgrimage centres in Poland is

national and patriotic symbolism. Its presence is historically conditioned and results from the strong connection between the national consciousness of Poles and Catholic piety.

It was noticed that the number and time and thematic scope of monuments or commemorative plaques related to historical events is diverse. The number and nature of the commemoration was related to the rank and history of the center of worship. Today, national content is still strongly present in pilgrimage centres in Poland. The organization of patriotic services and pilgrimages, as well as the foundation of objects with national symbolism in pilgrimage centres, is a dynamic process. Although their integrating role is no longer as strong as in the period of political repression, one can still notice relationships between religiousness and national consciousness, unheard of in other European countries.

**Author Contributions:** The contribution of all authors to the various stages of the article creation is equal. All authors have read and agreed to the published version of the manuscript.

**Funding:** Proofreading of this publication has been supported by a grant from the Priority Research Area (FutureSOC and Heritage) under the Strategic Program Excellence Initiative at Jagiellonian University; Research was financed by the project Heritage under the Strategic Program Excellence Initiative at Jagiellonian University, No. U1U/P01/NO/52.20, PI: dr Justyna Liro.

**Conflicts of Interest:** The authors declare no conflict of interest.

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
