# Peer review of "Symbolic Religious Landscape: Religious and Patriotic Symbolism in the Pilgrimage Centres in Poland"

_religions, doi:10.3390/rel14010033_

Round 1
Reviewer 1 Report
1) The article needs to be expanded to include more examples of the connection between religious and national symbolism. In this context, the specificity of this connection in Poland should be emphasized.
2) The article appropriately defines 3 aims, but aims 2 and 3 are intertwined and it is stated up front that there is a strong correlation between religious and national symbolism. I would see the pilgrimage centres here primarily as a means to a sense of national identity formation.
3) How was the sense of national identity assessed? This must be stated in the methodology.
4) A table with individual model places of pilgrimage (12) with their characteristics would be useful for this article.
5) I recommend to complete the current (after 2018) thematic literature.
6) The authors did not deal much with the historical changes of Poland's borders, which have changed considerably. In this context, it would be useful to mention the regional distribution of the model pilgrimage sites and consider including a map.
7) I understand that the authors have focused only on qualitative methods, but nevertheless, it would be suggested to carry out a quantitative analysis of the transformations of pilgrimage sites as well, mainly because it would better show, for example, the proportions of sacral objects intertwined with national symbolism.
8) I am surprised that the authors do not cite current research on the sacralization of John Paul II.
9) The paper also requires an extension of the final and synthetic findings.
Author Response
First, we would like to thank the Reviewer for his insightful and contributing comments. We have addressed them in the response below and they have been included in the manuscript. In addition, the article was checked in terms of language, style and editorial.
Reviewer's remarks:
1) The article needs to be expanded to include more examples of the connection between religious and national symbolism. In this context, the specificity of this connection in Poland should be emphasized.
Referring to this remark – papers on national and religious symbolism were citted at the beginning of chapter 2.1:
Historical and socio-cultural conditioning has a significant impact on the current network of pilgrimage centres in Poland and their symbolic religious landscape. The analysis of the presence of national symbolism in the Roman Catholic Church in Poland and religious symbolism in the ‘historical consciousness’ of Poles is present in the state of knowledge. Religious symbolism is analysed in the context of European values (e.g. MydÅ‚owska 2022) and as an important element of national identity (Mach 2007; Topidi 2019).
And also in chapter 2.2.
References to these publications have been added to the final list.
2) The article appropriately defines 3 aims, but aims 2 and 3 are intertwined and it is stated up front that there is a strong correlation between religious and national symbolism. I would see the pilgrimage centres here primarily as a means to a sense of national identity formation.
According to the Reviewer's comment, the second and third objectives were combined
3) How was the sense of national identity assessed? This must be stated in the methodology.
With regard to the remark, the following sentences have been added:
The presence of national symbols as well as plaques and monuments commemorating events from the history of Poland were assessed using qualitative methods - during the field inventory - by comparing the type and manner of commemoration. Their presence in a sacral building or in the sanctuary zone indicated the presence of a sense of national identity.
4) I recommend to complete the current (after 2018) thematic literature.
According to the Reviewer's comment, the thematic literature has been supplemented with the most up-to-date items (highlighted in red in the revised text of the manuscript).
5) The authors did not deal much with the historical changes of Poland's borders, which have changed considerably. In this context, it would be useful to mention the regional distribution of the model pilgrimage sites and consider including a map.
The article was supplemented with a map of the analysed pilgrimage centers in the context of changes in Poland's borders: Figure 1. Location of the analysed sanctuaries within the historical borders of Poland
6) I am surprised that the authors do not cite current research on the sacralization of John Paul II.
With regard to this remark, the article was supplemented with the following paragraph:
He was commemorated in very different ways in the public space. Therefore, there are streets, schools, hospitals, an airport named after him and about 800 monuments and 3 sanctuaries. The dynamics of its commemoration was so significant that there was talk of 'John-Paul-the-Second-isation' of cultural landscape in Poland (Przybylska, Sołjan 2015).
Reference to this publication has been added to the final list.
7) The paper also requires an extension of the final and synthetic findings.
According to the reviewer's comment, this part was expanded.
With regard to comments no. 4 and 7 in the original version of the review - they are very significant and contribute to reflection on this issue, the qualitative approach along with the empirical research that expands them and the tabular summaries are the result of our project, the results of which will be published in the next article , planned to be sent in February 2023. The volume range of the article as well as the structure and logic of this paper do not allow for including them here – in the opinion of the authors - and is the subject of consideration in a separate paper.
Reviewer 2 Report
The paper presents the specificity of Polish sanctuaries. The contained information may be particularly valuable for foreigners who do not know Polish historical, religious and cultural specifics. For Poles, this is not groundbreaking knowledge. Nevertheless, the paper collects and organizes selected concepts well. Sometimes he also embeds the knowledge of Polish sanctuaries in the European context. The following papers can assist in the
development of the topic
Stróżewski W. Wartość artystyczna i nadestetyczna, in: Sztuka i wartość , red. M. PoprzÄ™dzka , Warszawa 1986
Walczak R. 2005. Symbolika i wystrój Å›wiÄ…tyni chrzeÅ›cijanskiej, PoznaÅ„
Wawrzyniak W. 1996. Sacrum w architekturze, Wrocław
Author Response
First, thank you for reviewing. We refer to the remark below:
The paper presents the specificity of Polish sanctuaries. The contained information may be particularly valuable for foreigners who do not know Polish historical, religious and cultural specifics. For Poles, this is not groundbreaking knowledge. Nevertheless, the paper collects and organizes selected concepts well. Sometimes he also embeds the knowledge of Polish sanctuaries in the European context. The following papers can assist in the development of the topic
The articles proposed by the Reviewer are published only in Polish. We believe that they have a substantive value, however, quoting them in an English-language article is not justified, because they will be incomprehensible to foreign readers. In our opinion, the article adequately discusses the cultural conditions and the historical background of the presented issues. In our opinion, the articles to which we refer in the work are sufficient, fully cover the issues and have an international reach.
Reviewer 3 Report
This is a very good overview of pilgrimage sites in Poland, with lots of historical background.
The reviewer suggests that the authors do a careful check for minor spelling and punctuation errors before final publication, but these errors are few and mostly insignificant.
Author Response
First, we would like to thank the Reviewer.
This is a very good overview of pilgrimage sites in Poland, with lots of historical background. The reviewer suggests that the authors do a careful check for minor spelling and punctuation errors before final publication, but these errors are few and mostly insignificant.
The minor errors mentioned by the Reviewer have been corrected throughout the manuscript.